# GENOME-FACTORY: A LIBRARY FOR TUNING, DEPLOYING, AND INTERPRETING GENOMIC MODELS

## ABSTRACT

We introduce GENOME-FACTORY, the first integrated Python library for tuning, deploying, and interpreting genomic models. Our core contribution is to simplify and unify the workflow for genomic model development: data collection, model tuning, inference, benchmarking, and interpretability. For data collection, GENOME-FACTORY offers an automated pipeline to download genomic sequences and preprocess them. It also includes quality control like GC content normalization. For model tuning, GENOME-FACTORY supports three approaches: full-parameter, low-rank adaptation, and adapter-based fine-tuning. It is compatible with a wide range of genomic models. For inference, GENOME-FACTORY enables both embedding extraction and DNA sequence generation. For benchmarking, we include two existing benchmarks and provide a flexible interface for users to incorporate additional benchmarks. For interpretability, GENOME-FACTORY introduces the first open-source biological interpreter based on a sparse auto-encoder. This module disentangles embeddings into sparse, near-monosemantic latent units and links them to genomic features by regressing on external readouts. To improve accessibility, GENOME-FACTORY offers a zero-code command-line and a user-friendly web interface. We validate the utility of GENOME-FACTORY across three dimensions: (i) Compatibility with diverse models and fine-tuning methods; (ii) Benchmarking downstream performance using two open-source benchmarks; (iii) Biological interpretation of learned representations with DNABERT-2. These results highlight its end-to-end usability and practical value for real-world genomic analysis.

## 1 INTRODUCTION

We introduce GENOME-FACTORY, the first integrated Python library for tuning, deploying, and interpreting genomic foundation models (GFMs). GFMs have transformed biological research by enabling accurate and scalable solutions to key tasks, including epigenetic prediction (Gao et al., 2024), regulatory element discovery (Hwang et al., 2024), and species classification (Zhou et al., 2025b). These models learn from large-scale genomic data and support progress in personalized medicine, evolutionary biology, and functional genomics (Consens et al., 2025). Despite the potential of GFMs, their adoption in life sciences remains limited due to a fundamental gap between domain expertise and technical implementation. On one hand, engineers handle model training and deployment but often lack a biological context. Conversely, biologists design experiments and define scientific goals but lack expertise for large models. To address this, GENOME-FACTORY offers a unified, practical platform to bridge this gap and accelerate the broader use of GFMs in life science research.

While general-purpose language model fine-tuning frameworks such as LLaMA-Factory (Zheng et al., 2024) provide integrated tools, they do not address the unique requirements of genomics. Firstly, genomic data demands specialized handling, including support for domain-specific formats like FASTA, integration with repositories such as NCBI (Geer et al., 2010) for data acquisition, and domain-specific data preprocessing. Secondly, developers have built genomic models across diverse environments with heterogeneous dependencies. This lack of standardization makes it difficult to use models within a unified framework, and even harder to ensure compatibility with tools from the language model ecosystem. Thirdly, fine-tuning genomic models differs from language objectives: rather than instruction tuning or text generation, biological tasks often involve predicting variant effects, enhancer activity, or gene expression. These tasks require custom model adaptations and biology-informed loss functions aligned with real-world genomic objectives. Fourthly, evaluation further depends on domain-specific benchmarks, such as variant detection or regulatory site classifi-

cation (Zhou et al., 2024). These diverge from the text-based benchmarks used to assess language models. Finally, biological interpretability is central to the utility of GFMs for scientists, whereas it is not a focus of existing language model fine-tuning frameworks. As a result, the GFMs field still lacks a unified, user-friendly platform to support the full pipeline for tuning and deploying models.

To address this challenge, we introduce GENOME-FACTORY, the first unified Python library for fine-tuning, deploying, and interpreting genomic models. GENOME-FACTORY features six modular components. (i) **Genome Collector**: Retrieves genomic sequences from public repositories (e.g., NCBI (Geer et al., 2010)) and applies preprocessing such as GC content normalization and ambiguous base correction. It also includes task-specific dataset builders for histone modification, enhancer, and promoter classification, with automated region extraction and labeling. (ii) **Model Loader**: Supports diverse GFMs, including discriminative models—HyenaDNA (Nguyen et al., 2023), DNABERT-2 (Zhou et al., 2024), Caduceus (Schiff et al., 2024), and Nucleotide Transformer (Dalla-Torre et al., 2025)—as well as generative models such as EVO (Nguyen et al., 2024) and GenomeOcean (Zhou et al., 2025a). (iii) **Model Trainer**: Enables full-parameter fine-tuning and parameter-efficient methods such as low-rank adaptation (LoRA) (Hu et al., 2022) and adapter tuning (He et al., 2021). It applies to both classification and regression tasks. (iv) **Inference Engine**: Facilitates both embedding extraction and sequence generation. (v) **Benchmarker**: Provides two built-in, open-source genomic benchmarks and a plugin system for incorporating custom, domain-specific evaluation tasks and datasets. (vi) **Biological Interpreter**: Enhances model interpretability through a sparse auto encoder. It disentangles embeddings into near-monosemantic units and aligns them with genomic features via regression against external biological readouts. This is the first open-source tool to interpret the internal representations of GFMs. GENOME-FACTORY also offers user-friendly interfaces: a zero-code command-line interface (CLI) and an intuitive Gradio-based web-based user interface (WebUI) (Abid et al., 2019). These support both non-expert users and advanced developers in executing complex workflows with minimal computational and engineering effort.

In summary, we have the following three main contributions:

- We introduce GENOME-FACTORY, the first integrated Python framework to streamline the complete genomic model workflow. It integrates six components: (i) Genome Collector: comprehensive data collection and preprocessing (Section 3.1); (ii) Model Loader: broad support for diverse genomic models (Section 3.2); (iii) Model Trainer: an easy-to-use fine-tuning pipeline (Section 3.3); (iv) Inference Engine: efficient embedding extraction and sequence generation (Section 3.4); (v) Benchmarker: built-in genomic benchmarks and extensible evaluation plugins (Section 3.5); (vi) Biological Interpreter: model interpretability via sparse auto-encoder (Section 3.6).

- Beyond flexibility and ease of use, GENOME-FACTORY is the first unified framework to bring together diverse genomic models under a single interface. This enables seamless model comparison and assists users in selecting the most suitable model for a customized task. Notably, the Biological Interpreter is the first open-source tool to decode the internal representations of GFMs with a sparse auto-encoder. This provides deeper biological insights into model behavior.

- We validate the utility of GENOME-FACTORY across three dimensions: (i) Compatibility with diverse genomic foundation models and three widely used fine-tuning methods; (ii) Benchmarking downstream performance using two open-source benchmarks: Genome Understanding Evaluation (GUE) Benchmark (Zhou et al., 2024) and Genomic Benchmarks (Grešová et al., 2023); (iii) Biological interpretation of learned representations with DNABERT-2. These results clearly highlight its end-to-end usability and practical value for real-world genomic analysis.

**Organization.** Section 2 discusses related works on genomic foundation models and libraries for language models. Section 3 details the GENOME-FACTORY, including Genome Collector, Model Loader, Model Trainer, Inference Engine, Benchmarker, and Biological Interpreter. Section 4 presents the results of our experiments to evaluate GENOME-FACTORY's effectiveness.

## 2 RELATED WORK

In the following, we first discuss the genomic foundation models in Section 2.1. Next, we discuss the existing libraries for natural language models in Section 2.2.

### 2.1 GENOMIC FOUNDATION MODELS

Several powerful genomic foundation models have recently emerged to decode the complex language of DNA. For the discriminative model domain, HyenaDNA (Nguyen et al., 2023) employs implicit

convolution layers. This enables million-token contexts at single-nucleotide resolution to capture complex regulatory interactions and support in-context species classification. DNABERT-2 (Zhou et al., 2024) employs byte pair encoding and a refined transformer architecture. It improves tasks such as epigenomic mark prediction and transcription factor binding. Caduceus (Schiff et al., 2024) leverages the Mamba architecture (Gu and Dao, 2024) with reverse-complement equivariance to improve long-range variant-effect prediction. Nucleotide Transformer (Dalla-Torre et al., 2025) scales to 2.5B parameters with 6-mer tokenization. It achieves strong performance in chromatin-feature prediction and functional variant prioritization. For the generative model domain, Evo (Nguyen et al., 2024) introduces a 7B-parameter generative model to extend beyond embedding extraction. This enables genome-scale predictions across biomolecular modalities. GenomeOcean (Zhou et al., 2025a) further improves generative efficiency for metagenomic sequence synthesis. It advances applications in synthetic biology. However, diverse models use a wide range of environments with heterogeneous dependencies and configurations. This lack of standardization makes it difficult to load, fine-tune, or compare models. Such fragmentation increases the technical burden on users and limits the accessibility of genomic models. To address this, we introduce GENOME-FACTORY, the first Python framework to unify and streamline the end-to-end genomic model workflow.

## 2.2 LIBRARIES FOR LANGUAGE MODELS

In parallel, the language community has developed numerous frameworks to streamline the adaptation and fine-tuning of language models. These toolkits target different stages of the language model lifecycle. For example, LLaMA-Adapter (Zhang et al., 2024) improves fine-tuning efficiency, while GPT4All (Anand et al., 2023) enables practical model training and efficient inference on widely available consumer-grade hardware. Other frameworks address specific training challenges or model architectures: Colossal-AI (Li et al., 2023) introduces advanced parallelism strategies for efficient large-scale distributed training, FastChat (Zheng et al., 2023) provides specialized tools for building dialogue agents, and Open-Instruct (Wang et al., 2023) standardizes methodologies for instruction tuning. Flexibility, and domain specialization have also emerged as key priorities. LitGPT (Saroufim et al., 2025) adopts a modular design to support diverse generative model training paradigms, while LMFlow (Diao et al., 2024) helps researchers train language models for specific domains. Finally, LLaMA-Factory (Zheng et al., 2024) further unifies this ecosystem by integrating multiple efficient fine-tuning techniques into a single, comprehensive toolkit. However, these frameworks do not translate to the genomic domain. Genomic models require specialized data formats, biology-informed objectives, domain-specific benchmarks, and meaningful biological insights. GENOME-FACTORY fills this gap with tools tailored for genomic models. It supports data collection, model tuning, inference, benchmarking, and biological interpretation. Notably, the Biological Interpreter is the first open-source tool to decode the hidden internal representations of genomic models.

## 3 GENOME-FACTORY FRAMEWORK

GENOME-FACTORY comprises six core modules to unify the workflow for genomic models. The Genome Collector (Section 3.1) simplifies data retrieval (e.g., from NCBI (Geer et al., 2010)) and integrates preprocessing pipelines. The preprocessing includes the quality control steps, such as sequence-length filtering, GC content normalization, and correction of ambiguous bases. It also supports task-specific dataset builders for histone modification, enhancer, and promoter classification, with automated region extraction, chromosome-name harmonization, and labeling. Model Loader (Section 3.2) and Model Trainer (Section 3.3) handle model loading and fine-tuning. They support both full and parameter-efficient methods, such as LoRA and adapters for classification and regression tasks. The Inference Engine (Section 3.4) enables embedding extraction and sequence generation with pre-trained genomic foundation models. The Benchmarker (Section 3.5) provides essential tools and datasets for rigorous model evaluation and comparison across tasks. The Biological Interpreter (Section 3.6) delivers interpretability via a sparse auto-encoder. It learns near-monosemantic latent units from model embeddings and links them to interpretable genomic features (e.g., motif presence, sequence length) through regression on external biological readouts. Users can access GENOME-FACTORY through a zero-code command-line interface or an interactive web-based user interface (Section 3.7). This design supports both flexibility and scalability for diverse applications. We provide an overview of GENOME-FACTORY in Figure 1, and **detailed usage examples** in Appendix E.

### 3.1 GENOME COLLECTOR

The Genome Collector manages the upstream data pipeline for genomic models. It automates the fetching, transformation, and validation of genomic data. This extends beyond basic sequence

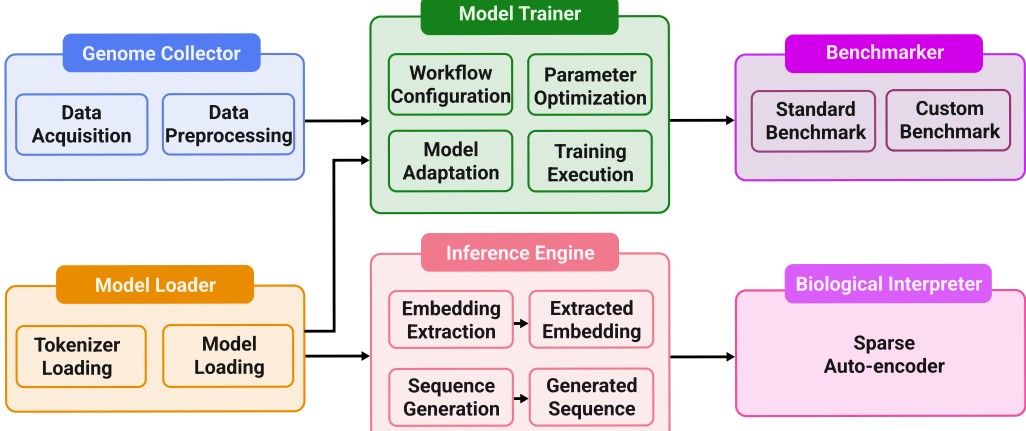

Figure 1: **Overview of GENOME-FACTORY.** The framework consists of six components. **Genome Collector** acquires genomic sequences from public repositories and performs preprocessing (e.g., GC normalization, ambiguous base correction). **Model Loader** supports major genomic models (e.g., HyenaDNA, DNABERT-2, Caduceus, Nucleotide Transformer, EVO and GenomeOcean) and their tokenizers. **Model Trainer** configures workflows, adapts models to classification or regression tasks, and executes training with full fine-tuning or parameter-efficient methods (LoRA, adapters). **Inference Engine** enables embedding extraction and sequence generation. **Benchmarker** provides standard benchmarks and allows integration of custom evaluation tasks. **Biological Interpreter** enhances interpretability through sparse auto-encoders.

downloads to include regulatory and epigenetic annotations. By standardizing dataset construction across diverse biological tasks, it ensures high-quality, task-ready inputs for the model.

**Data Acquisition.** The Genome Collector offers multiple flexible pipelines for downloading and preparing diverse DNA sequence datasets. Beyond basic genome-wide retrieval from public repositories such as NCBI, the framework supports three task-driven acquisition modes. (i) It constructs region-based datasets by identifying signal-enriched locations, such as high-coverage intervals from genome-wide profiles (e.g., histone modification enrichment). (ii) It builds binary classification datasets by separating annotated functional regions from background sequences, such as distinguishing regulatory elements from random regions (e.g., enhancers). (iii) It samples region-versus-background pairs by comparing annotated start sites to non-start regions of matched length (e.g., promoters). Each pipeline handles file downloading, genome indexing, sequence extraction, chromosome name harmonization, and sequence quality filtering. A unified interface lets users choose the task type and generate corresponding datasets with minimal manual intervention.

**Data Preprocessing.** Each acquisition mode applies task-specific parsing and transformation logic. For species-level classification, the system samples fixed-length DNA fragments from genomes and assigns species labels. The histone modification pipeline extracts gene-body sequences aligned to signal peaks and binarizes them into high/low classes based on enrichment thresholds. The enhancer and promoter pipelines define positive regions from regulatory annotations and sample negatives from non-overlapping regions. Genome Collector saves each dataset in a standardized format with DNA sequences and corresponding labels, and partitions it into training, validation, and testing sets.

**Quality Control and Data Cleaning.** To ensure reliable and robust downstream performance, Genome Collector applies a multi-stage quality control protocol. Initial filters enforce several basic constraints on sequence length, GC content, and the overall proportion of ambiguous nucleotides. Further statistical quality control removes outliers using three strategies: (i) filtering sequences with excessive ambiguous bases; (ii) applying chi-square tests to detect compositional biases relative to expected nucleotide distributions; and (iii) removing rare compositional profiles below a predefined frequency threshold. Quality control steps confirm that the cleaned datasets exhibit balanced GC content and sequence length distributions. These safeguards collectively ensure the final datasets have valid biological structure and robust statistical properties for training genomic models.

## 3.2 MODEL LOADER

The Model Loader handles the initial phase of inference or fine-tuning by loading the selected genomic model and corresponding tokenizer. It leverages the Hugging Face Transformers (Wolf et al., 2019) to support a broad range of powerful models, including HyenaDNA (Nguyen et al.,

2023), DNABERT-2 (Zhou et al., 2024), Caduceus (Schiff et al., 2024), Nucleotide Transformer (Dalla-Torre et al., 2025)), EVO (Nguyen et al., 2024) and GenomeOcean (Zhou et al., 2025a).

**Tokenizer Loader.** The system automatically loads the appropriate tokenizer for the selected genomic model using the Hugging Face tokenizer API. This ensures accurate and consistent encoding of input DNA sequences and full compatibility with the model's input format.

**Model Architecture and Checkpoint Loader.** After initializing the tokenizer, the system loads the model with either pretrained checkpoints or random parameters. It configures the architecture based on the selected fine-tuning method. For full-parameter or LoRA fine-tuning, it loads the model and attaches a task-specific classification or regression head. For adapter fine-tuning, it freezes the base model's weights and inserts an external adapter module for task-specific training.

## 3.3 MODEL TRAINER

Adapting large genomic models to downstream tasks poses significant computational challenges due to their size and parameter count. The Model Trainer manages configuration, launches fine-tuning jobs, and monitors training. To address scalability, it combines parameter-efficient strategies with optimization techniques. The module includes four components: workflow configuration, parameter optimization, model adaptation, and training execution. Table 1 summarizes the compatibility of training strategies and system-level optimizations.

Table 1: **Featured tuning techniques.**

|  | Full | LoRA | Adapter |
|---|:---:|:---:|:---:|
| **Mixed precision** | ✓ | ✓ | ✓ |
| **Flash attention** | ✓ | ✓ | ✓ |
| **Gradient accumulation** | ✓ | ✓ | ✓ |

**Workflow Configuration.** The system automatically reads user inputs and builds task-specific configurations for a classification or regression task. Custom parsers validate hyperparameters and construct training pipelines based on the selected fine-tuning strategy. The Hugging Face Trainer manages core training and distributed execution to ensure reliability, consistency, and scalability.

**Parameter Optimization.** GENOME-FACTORY offers three fine-tuning strategies to effectively balance resource efficiency and model adaptability: (i) Full-parameter fine-tuning updates all model parameters. It maximizes task adaptation but incurs a high computational cost. (ii) Low-rank adaptation (LoRA) (Hu et al., 2022) freezes the base model and introduces trainable low-rank matrices in attention or feed-forward layers. It reduces memory and training time while preserving performance. (iii) Adapter tuning (He et al., 2021) adds a lightweight neural module (e.g., multilayer perceptrons (Popescu et al., 2009) or convolutional neural network (O'shea and Nash, 2015)) after the frozen base model. It only updates the adapter parameters. All methods support customizable hyperparameters, including learning rate, dropout, weight decay, LoRA rank, and scaling factor.

**Model Adaptation.** GENOME-FACTORY adjusts model architectures based on the task. For classification, it appends activation functions such as Softmax to the output layer and applies cross-entropy loss (De Boer et al., 2005). For regression, it sets up continuous-valued outputs and uses mean squared error loss (Schluchter, 2005). The system handles variable-length sequences using dynamic padding or truncation to maintain compatibility with model input requirements.

**Training Execution.** GENOME-FACTORY seamlessly integrates the following performance optimizations: (i) Precision control supports full, FP16, and BF16 formats to reduce memory usage and speed up training on compatible hardware. (ii) Flash attention (Dao et al., 2022) accelerates attention computation and minimizes memory by avoiding explicit intermediate matrices. (iii) Gradient accumulation simulates large batch sizes, and learning rate scheduling improves convergence stability. It also applies gradient clipping to prevent exploding gradients. Furthermore, it supports multi-GPU training via distributed data parallel (Li et al., 2020). During training, it logs evaluation metrics at configurable intervals. For classification tasks, it tracks accuracy, F1 score, precision, recall, and Matthews correlation coefficient. For regression, it records mean squared error and mean absolute error. The system saves periodic checkpoints and retains the best-performing one based on validation metrics. This enables flexible training resumption or model deployment.

## 3.4 INFERENCE ENGINE

The Inference Engine offers a unified interface for effectively applying genomic foundation models to two key tasks: sequence embedding extraction and DNA sequence generation.

**Embedding Extraction.** This component processes input DNA sequences, runs the model in evaluation mode, and extracts the final hidden state as the sequence embedding. These embeddings support downstream tasks such as classification, regression, clustering, and visualization.

**Sequence Generation.** This component enables compatible models to flexibly generate diverse novel DNA sequences from user-provided prompts. It supports applications such as in silico sequence variation, synthetic data augmentation, and functional sequence exploration.

### 3.5 BENCHMARKER

The Benchmarker module provides essential tools for evaluating genomic foundation models on both classification and regression tasks. It supports standardized benchmark integration, flexible plugins for custom domain-specific evaluation tasks, and automated performance evaluation.

**Incorporating Benchmarks.** GENOME-FACTORY includes two benchmark suites: the Genome Understanding Evaluation (GUE) Benchmark (Zhou et al., 2024) and Genomic Benchmarks (Grešová et al., 2023). It also supports plugins for integrating custom, domain-specific evaluation tasks. To use a custom benchmark, users format their data according to the Model Trainer's input schema. For classification, the dataset should include three CSV files (training, validation, and testing). Each contains two columns: one for DNA sequences and one for integer labels. For regression, users follow the same structure but replace the label column with continuous values.

**Evaluating Models.** GENOME-FACTORY runs the selected model on the benchmark dataset and records detailed task-specific metrics. For classification, it computes accuracy, F1 score, precision, recall, and Matthews correlation coefficient. For regression, it computes mean squared error and mean absolute error. The system logs all results in a structured JSON file. This allows users to easily compare model performance across tasks, datasets, or different training strategies.

### 3.6 BIOLOGICAL INTERPRETER

The Biological Interpreter module enables meaningful interpretation of genomic foundation models by explicitly linking internal model representations to biological features. This supports novel hypothesis generation and provides deeper insight into what the model has actually learned.

**Sparse Auto-encoder.** GENOME-FACTORY uses a sparse auto-encoder to disentangle latent embeddings from genomic models. The workflow involves three stages: (i) Sequence embedding extraction: The system embeds input DNA sequences with a pretrained genomic model, such as GenomeOcean. (ii) Sparse auto-encoder training: It trains a sparse auto-encoder on these embeddings with a reconstruction loss. The training enforces sparsity so that only a small subset of latent units activate per sequence. This encourages each unit to capture a distinct, near-monosemantic genomic feature. (iii) Regression to external readouts: The system fits regression models between the sparse latent units and external biological readouts (e.g., sequence length, motif presence, or experimental measurements) to associate individual neurons with interpretable molecular features.

### 3.7 COMMAND-LINE AND WEB-BASED USER INTERFACE

GENOME-FACTORY offers both a command-line interface (CLI) and a web-based user interface (WebUI) to accommodate a range of user preferences and expertise levels.

**Command-line Interface.** Users access the command-line interface through a single convenient entry point, `genomefactory-cli`, and define configuration-first workflows with YAML. Users specify tasks, datasets, models, and training or inference settings through compact configuration files. The command-line interface supports four core functionalities: (i) data acquisition and preprocessing with task-specific dataset builders and automated quality control; (ii) model training using full fine-tuning or parameter-efficient methods; (iii) inference for embedding extraction and sequence generation; and (iv) interpretability using the sparse auto-encoder–based Biological Interpreter. The system saves all metrics and artifacts for downstream evaluation and reproducibility.

**Web-based User Interface.** The WebUI uses Gradio (Abid et al., 2019) to complement the CLI and gives users an intuitive, code-free way to conveniently access all core GENOME-FACTORY features. Users configure data, models, and tasks for training, inference, benchmarking, and interpretability through a clear graphical layout. The interface shows relevant parameters based on the selected method, applies sensible defaults, and organizes workflows into task-specific tabs. Users launch tasks with a single click and view logs and results in real time within the browser. By hiding the underlying code, the WebUI lets researchers interact with models through a graphical interface.

## 4 EMPIRICAL STUDY

We comprehensively demonstrate the effectiveness of GENOME-FACTORY through three key dimensions. Each dimension highlights the partial capabilities of six integrated modules: (i) Fine-tuning

Table 2: **Fine-tuning efficiency with different methods in GENOME-FACTORY.** We report the number of trainable parameters, peak GPU memory usage, and throughput (thousands of tokens per second). "—" marks settings we did not evaluate due to computational constraints (e.g., EVO with full fine-tuning). We conduct all experiments on a single NVIDIA H100 (80GB). Due to the large size of EVO (7B) and its computational demands, we exclude full-parameter fine-tuning for this model. "Full" denotes full-parameter fine-tuning, "LoRA" denotes low-rank adaptation, and "Adapter" denotes adapter-based tuning.

| Method | HyenaDNA-160k | | | DNABERT2 | | | Caduceus-131k | | |
|---|---|---|---|---|---|---|---|---|---|
| | Trainable params | Mem (GB) | Throughput (KTok/s) | Trainable params | Mem (GB) | Throughput (KTok/s) | Trainable params | Mem (GB) | Throughput (KTok/s) |
| Full | 6.55M | 5.75 | 381.48 | 117.08M | 7.30 | 40.70 | 7.73M | 6.89 | 143.55 |
| LoRA | 0.10M | 4.45 | 446.05 | 1.49M | 6.18 | 44.07 | 0.81M | 1.85 | 394.30 |
| Adapter | 0.07M | 1.84 | 1113.32 | 0.20M | 2.09 | 124.30 | 0.07M | 1.85 | 467.72 |

| Method | Nucleotide Transformer-500M | | | EVO-1-131k | | | GenomeOcean-100M | | |
|---|---|---|---|---|---|---|---|---|---|
| | Trainable params | Mem (GB) | Throughput (KTok/s) | Trainable params | Mem (GB) | Throughput (KTok/s) | Trainable params | Mem (GB) | Throughput (KTok/s) |
| Full | 480.45M | 18.94 | 12.08 | — | — | — | 116.42M | 7.39 | 45.28 |
| LoRA | 4.46M | 15.95 | 14.49 | 0.39M | 84.85 | 6.02 | 1.70M | 7.00 | 45.88 |
| Adapter | 0.33M | 4.10 | 41.28 | 1.05M | 14.74 | 13.03 | 0.20M | 2.74 | 137.48 |

compatibility across diverse models (Section 4.1): This dimension evaluates the compatibility of three fine-tuning strategies: full, LoRA, and adapter tuning. It covers a range of models. It showcases the functionality of the Model Loader, Model Trainer, and Inference Engine. (ii) Benchmarking downstream performance (Section 4.2): By benchmarking different models on standardized downstream tasks, we analyze their trade-offs between accuracy and computational efficiency. This dimension emphasizes the role of the Benchmarker module. (iii) Biological interpretation of learned representations with DNABERT-2 (Section 4.3): We explore how model embeddings capture biological signals, such as correlations with sequence length, to assess interpretability. This dimension demonstrates the capability of the Genome Collector and Biological Interpreter modules.

## 4.1 FINE-TUNING OF DIVERSE MODELS

We evaluate the six genomic foundation models from the Model Loader using all three fine-tuning strategies from the Model Trainer. Notably, the adapter-based method requires extracting base model embeddings before training. This highlights the functionality of the Inference Engine.

**Experimental Setup.** We systematically evaluate the training efficiency of full fine-tuning, LoRA, and adapter tuning using the COVID variant prediction task from the GUE benchmark (Zhou et al., 2024). This task involves classifying sequences into one of nine labels. We test six models: HyenaDNA-160k, DNABERT-2, Caduceus-131k, Nucleotide Transformer-500M, EVO-1-131k, and GenomeOcean-100M. Please see Appendix C.1 for more experimental details.

**Results.** We present a detailed quantitative comparison of training efficiency across three different fine-tuning strategies in Table 2. The table reports the number of trainable parameters, peak GPU memory consumption, and throughput measured in thousands of tokens per second. Among all available methods, adapter tuning consistently shows the highest efficiency. For example, the DNABERT-2 adapter uses only 0.2 million trainable parameters, consumes 2.09 gigabytes of memory, and reaches a throughput of 124,000 tokens per second. In contrast, full fine-tuning for the same model updates 117 million parameters, uses 7.30 gigabytes of memory, and processes only 41,000 tokens per second. The adapter module allows users to adjust its internal structure to meet specific hardware or speed constraints. LoRA also offers strong efficiency gains. For DNABERT-2, it reduces the parameter count to 1.49 million and improves throughput to 44,000 tokens per second while reducing memory usage compared to full fine-tuning. Full fine-tuning remains the most expensive. It requires updating all model parameters, consumes the most GPU memory, and achieves the lowest throughput. Due to its extreme cost, we exclude full fine-tuning for the 7B-parameter EVO model.

To assess scalability, we compare LoRA and full fine-tuning across different model sizes in Table 3. For the Nucleotide Transformer, scaling from 500 million to 2.5 billion parameters lowers the LoRA parameter ratio from 0.9% to 0.5%, increases memory savings from 16% to 32%, and boosts throughput from 1.20 to 1.39 times compared to full tuning. A similar trend appears in GenomeOcean. Scaling from 100 million to 500 million parameters decreases the parameter ratio from 1.5% to 0.7%, increases memory savings from 5% to 21%, and raises throughput from 1.01 to 1.27 times. These

Table 3: **Fine-tuning efficiency across different model scales.** We report experimental results to clearly illustrate the scalability of LoRA's efficiency with Nucleotide Transformer (500M/2.5B) and GenomeOcean (100M/500M) as representative examples. We conduct all experiments on a single NVIDIA H100 (80GB). "Full" denotes full-parameter fine-tuning, and "LoRA" denotes low-rank adaptation.

| Model | Method | Trainable Params | Peak Mem (GB) | Throughput (KTok/s) |
|---|---|---|---|---|
| **Nucleotide Transformer-500M** | **Full** | 480.45 M | 18.94 | 12.08 |
| | **LoRA** | 4.46 M | 15.95 | 14.49 |
| **Nucleotide Transformer-2.5B** | **Full** | 2.54 B | 63.95 | 2.40 |
| | **LoRA** | 11.86 M | 43.33 | 3.34 |
| **GenomeOcean-100M** | **Full** | 116.42 M | 7.39 | 45.28 |
| | **LoRA** | 1.70 M | 7.00 | 45.88 |
| **GenomeOcean-500M** | **Full** | 534.83 M | 19.58 | 11.85 |
| | **LoRA** | 3.97 M | 15.48 | 15.07 |

results show that LoRA becomes effective as model size grows, making it a strong choice for adapting large genomic models within GENOME-FACTORY. These findings align with expectations and demonstrate the functionality of our Model Loader, Model Trainer, and Inference Engine modules.

## 4.2 BENCHMARKING DIFFERENT MODELS

We comprehensively benchmark six genomic foundation models with three different fine-tuning strategies across two benchmark suites. This experiment highlights the role of the Benchmarker module by comparing model performance and tuning methods on standardized downstream tasks.

**Experimental Setup.** We evaluate model performance on tasks from two widely used sources: the GUE benchmark (Zhou et al., 2024) and Genomic Benchmarks (Grešová et al., 2023). For most benchmark tasks, we report reliable test set performance with the Matthews correlation coefficient. For the COVID variant prediction task, we follow the benchmark protocol and report the F1 score. We evaluate the same six models in Section 4.1: HyenaDNA-160k, DNABERT-2, Caduceus-131k, Nucleotide Transformer-500M, EVO-1-131k, and GenomeOcean-100M. For each experiment, we consistently use random seeds 14, 28, and 42, and report the mean score along with the standard deviation. Please see Appendix C.2 for more experimental details.

Table 4: **Benchmark across models and tuning methods.** We report results on both the GUE benchmark and Genomic Benchmarks. "—" marks settings we did not evaluate due to computational constraints. We exclude EVO (7B) from full fine-tuning due to the high computation cost. "Full" denotes full-parameter fine-tuning, "LoRA" denotes low-rank adaptation, and "Adapter" denotes adapter-based tuning.

| Model | GUE | | | Genomic Benchmarks | | |
|---|---|---|---|---|---|---|
| | **Full** | **LoRA** | **Adapter** | **Full** | **LoRA** | **Adapter** |
| **HyenaDNA-160k** | $59.91 \pm 0.22$ | $50.95 \pm 0.23$ | $25.00 \pm 0.37$ | $66.71 \pm 0.18$ | $61.26 \pm 0.29$ | $36.90 \pm 0.42$ |
| **DNABERT-2** | $65.25 \pm 0.25$ | $48.39 \pm 0.11$ | $40.19 \pm 0.17$ | $\mathbf{71.97 \pm 0.20}$ | $64.58 \pm 0.25$ | $48.96 \pm 0.31$ |
| **Caduceus-131k** | $50.07 \pm 1.61$ | $34.70 \pm 0.19$ | $38.61 \pm 0.33$ | $65.38 \pm 0.22$ | $42.10 \pm 0.36$ | $47.74 \pm 0.27$ |
| **Nucleotide Transformer-500M** | $57.63 \pm 0.26$ | $52.96 \pm 0.13$ | $32.01 \pm 0.48$ | $67.99 \pm 0.24$ | $64.68 \pm 0.30$ | $43.95 \pm 0.28$ |
| **EVO-1-131k** | — | $44.97 \pm 0.79$ | $30.08 \pm 0.22$ | — | $51.33 \pm 0.41$ | $33.69 \pm 0.34$ |
| **GenomeOcean-100M** | $\mathbf{65.52 \pm 0.57}$ | $59.35 \pm 0.23$ | $\mathbf{46.65 \pm 0.04}$ | $68.26 \pm 0.26$ | $\mathbf{66.28 \pm 0.19}$ | $\mathbf{53.60 \pm 0.21}$ |

**Results.** We show results on downstream tasks in Table 4. Due to the large size of the EVO model (7B parameters) and its high computational cost, we omit full fine-tuning results for this model. We report the averaged Matthews correlation coefficient across all evaluated datasets. Except for the COVID subset of the GUE benchmark, we follow the original protocol and use the macro-averaged F1 score. The overall GUE score is computed by averaging this F1 value with the correlation coefficients from the remaining tasks. We observe that GenomeOcean-100M achieves the strongest performance under full fine-tuning on the GUE, while DNABERT-2 performs best on the Genomic Benchmarks. Parameter-efficient methods remain competitive. For example, LoRA sometimes matches full fine-tuning, as seen with GenomeOcean-100M on Genomic Benchmarks (68.26 vs. 66.28). While Adapter method underperforms LoRA in most cases, it narrows the gap and even surpasses LoRA for Caduceus. This suggests that a task-aware adapter improves performance.

We visualize the trade-offs between predictive performance and computational efficiency for DNABERT-2, HyenaDNA, and Nucleotide Transformer in Figure 2 under three fine-tuning strategies: full fine-tuning, low-rank adaptation, and adapter-based methods. The figure reports memory usage, training throughput, and GUE scores. Adapter tuning reduces memory usage and boosts throughput

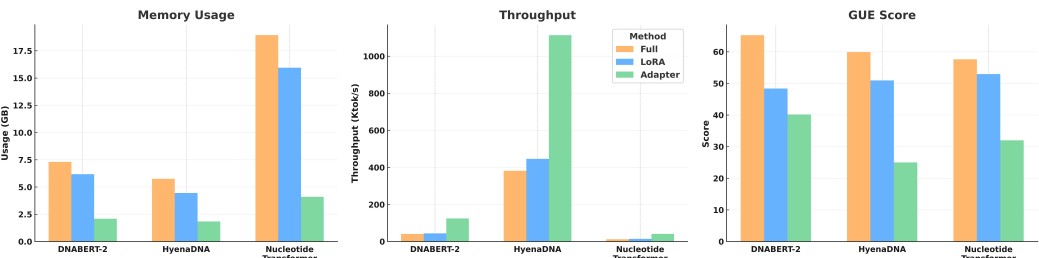

Figure 2: **Trade-off between tuning efficiency and performance.** The figure shows memory usage in gigabytes (GB), throughput in kilotokens per second (KTok/s), and averaged scores on the GUE benchmark for three models: DNABERT-2, HyenaDNA-160k, and Nucleotide Transformer-500M. We report results for full-tuning (Full), low-rank adaptation (LoRA), and adapter-based fine-tuning (Adapter). The results highlight the trade-offs between resource efficiency and predictive performance.

across all models, though it sacrifices some predictive performance. In contrast, full fine-tuning and low-rank adaptation yield higher GUE scores but require more compute. This comparison highlights the flexible trade-off space between efficiency and accuracy with GENOME-FACTORY. Overall, these results demonstrate both the functionality and practical utility of the Benchmarker module.

### 4.3 BIOLOGICAL INTERPRETATION OF GENOMIC MODELS

We clearly demonstrate the capabilities of the Genome Collector and Biological Interpreter modules by thoroughly analyzing model embeddings for interpretability.

**Experimental Setup.** We use the Genome Collector to automate comprehensive genomic data preparation. The Data Acquisition component downloads genome sequences for two organisms from NCBI: Arabidopsis thaliana and Bos taurus. We save all downloaded files in a unified directory. After acquisition, the Data Preprocessing component segments each genome into 1,000 sequences of random length between 500 and 1,000 base pairs. This yields a total of 2,000 segments. We then apply standard quality control procedures, including GC content correction and removal of ambiguous bases. Finally, we construct a labeled dataset by assigning each sequence a single label corresponding to its sequence length. To explore representation learning, we train a sparse autoencoder with a hidden dimension of 4,096 using both the input sequences and their reconstruction losses. After training, we use the hidden states of the autoencoder to perform regression on the sequence length labels. This clearly identifies which latent features are associated with sequence length.

**Results.** We find that the 382nd, 519th, and 3519th units exhibit strong correlations with the biological attribute of sequence length. This enables detailed biological interpretation at the individual neuron level. It allows researchers to formulate mechanistic hypotheses about what the model has learned. This is the first open-source system that bridges the gap between black-box genomic foundation models and interpretable biological insights in the genomic domain.

## 5 CONCLUSION AND FUTURE WORK

We introduce GENOME-FACTORY, the first unified and modular Python framework for streamlining the tuning, deployment, and interpretation of genomic models. GENOME-FACTORY offers an end-to-end pipeline that integrates six key components: Genome Collector for acquiring and preprocessing data; Model Loader for accessing genomic models; Model Trainer for fine-tuning models tailored to specific downstream tasks; Inference Engine for embedding extraction and sequence generation; Benchmarker for evaluating model performance; and Biological Interpreter for interpretability via sparse autoencoders. It supports various genomic models and enhances accessibility through a zero-code command-line interface and an intuitive web-based user interface. Our experiments demonstrate the utility of GENOME-FACTORY on key genomic downstream tasks using multiple training methods across diverse model architectures. These results highlight its end-to-end usability and practical value for real-world genomic analysis. GENOME-FACTORY lowers the technical barrier to using large-scale models in genomics research. It makes these powerful tools more accessible to the broader research community. Future work will involve tracking state-of-the-art genomic models and updating our repertoire. We also plan to incorporate novel training strategies, broaden the range of integrated genomic tasks and benchmarks to accelerate biological discovery, and further advance the Biological Interpreter to provide richer genomic interpretability. We discuss the limitations and broader impact in Appendix B, and provide detailed usage examples of GENOME-FACTORY in Appendix E.

## REPRODUCIBILITY STATEMENT

We ensure reproducibility through the complete code and configuration files provided in the supplementary materials. All experiments use fixed random seeds. These resources enable full replication of our results across tasks, models, and training methods.

## ETHIC STATEMENT

This paper does not involve human subjects, personally identifiable data, or sensitive applications. We do not foresee direct ethical risks. We follow the ICLR Code of Ethics and affirm that all aspects of this research comply with the principles of fairness, transparency, and integrity.

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

# Supplementary Material

## A  LLM USAGE ACKNOWLEDGEMENT

Large language models (LLMs) are used in this project as general-purpose assistive tools to support writing and coding tasks. We detail their specific roles below.

- **Manuscript Writing:** We use **GPT-4o** to polish the manuscript, including grammar correction, phrasing refinement, and improving clarity throughout the text.

- **Code Development:** We use **GPT-4o** for debugging, code generation, and resolving implementation issues during the library development.

All LLM-assisted content is reviewed and curated by the authors. We take full responsibility for the final codebase and manuscript.

## B  LIMITATIONS AND BROADER IMPACT

We provide the limitations and broader impact here.

**Limitations.**  We have the following three limitations:

- Due to the large size of EVO (7B) and the associated compute cost, we do not consider full fine-tuning for EVO, but we plan to include it in the future.

- We conduct all experiments with a unified pipeline and have not incorporated traditional computational biology techniques (e.g., PWM scanning (Ambrosini et al., 2018), HMM-based motif finding (Qin et al., 2010)). We will add these classical methods and compare them with the genomic model-based approach within a unified evaluation framework.

- For model fine-tuning, we include three main methods. In the future, we plan to incorporate more advanced training and computational acceleration techniques, including quantization (Egiazarian et al., 2024) and the liger kernel (Hsu et al., 2024).

**Broader Impact.**  GENOME-FACTORY lowers the barrier to advanced genomic modeling. It will accelerate research in personalized medicine and evolutionary biology while cutting computational costs and energy use. By promoting open, reproducible workflows, it fosters responsible innovation. Users must adhere to biosecurity and ethical guidelines when generating sequences.

## C    EXPERIMENTAL DETAILS

In this section, we provide the experimental details for Section 4.1 and Section 4.2.

### C.1    FINE-TUNING OF DIVERSE MODELS

For all experiments, we use AdamW with full-precision (FP32) updates, a fixed learning rate of $3 \times 10^{-5}$, batch size 32, and default model settings unless noted. For LoRA tuning, we set the rank to $r = 8$ and the scaling factor to $\alpha = 32$. For most models (DNABERT-2, Caduceus, Nucleotide Transformer, GenomeOcean), we apply LoRA to all feed-forward layers. For HyenaDNA, due to its specialized architecture, we apply LoRA only to the input/output projection layers within the Hyena blocks. For EVO, given the large size of its feed-forward layers, we target only the key, query, and value matrices in attention blocks. The adapter tuning keeps all base model parameters frozen and adds a trainable multilayer perceptron adapter with a single hidden layer of size 256. We conduct all experiments on a single NVIDIA H100 80GB GPU.

### C.2    BENCHMARKING DIFFERENT MODELS

We fine-tune all models with a learning rate of $3.0 \times 10^{-5}$ and the AdamW optimizer in full-precision (FP32) mode. For Caduceus, we use a higher learning rate of $1.0 \times 10^{-3}$ to ensure training stability. For LoRA tuning, we use rank $r = 8$ and scaling factor $\alpha = 32$. Similar to Appendix C.1, we apply LoRA to all feed-forward layers in DNABERT-2, Caduceus, Nucleotide Transformer, and GenomeOcean. For HyenaDNA, we apply LoRA only to the input and output projection layers within the Hyena blocks. For the EVO model, we target the key, query, and value matrices in each attention block, due to the large size of feed-forward layer parameters. The adapter tuning freezes all base models and appends a trainable multilayer perceptron adapter with a single hidden layer of size 256. We perform training on a single NVIDIA H100 80GB GPU for most models with a batch size of 32. For Nucleotide Transformer-500M, due to its larger memory footprint during full-parameter tuning, we fine-tune it with distributed data parallel across two NVIDIA H100 80GB GPUs. This also demonstrates the functionality of our distributed training.

## D    DETAILS OF SUPPORTED MODELS

The following tables show GENOME-FACTORY 's list of supported models.

Table 5: **Supported models in GENOME-FACTORY with their available variants.** Models vary by either parameter size (e.g., GenomeOcean: 100M/500M/4B) or input sequence length (e.g., Hyenadna: 1K to 1M). "Variant Type" specifies the axis of variation, and "Variants" lists the available options.

| Model | Variant Type | Variants |
|---|---|---|
| **Hyenadna** | Sequence Length | 1K/16K/32K/160K/450K/1M |
| **DNABERT-2** | Parameter Size | 117M |
| **Caduceus** | Sequence Length | 1K/131K |
| **Nucleotide Transformer** | Parameter Size | 50M/100M/250M/500M/1B/2.5B |
| **EVO** | Sequence Length | 8K/131K |
| **GenomeOcean** | Parameter Size | 100M/500M/4B |

# E  USAGE EXAMPLES OF GENOME-FACTORY

We provide usage examples of GENOME-FACTORY through two interfaces: the command-line interface (CLI) and the web-based interface (WebUI). We begin with detailed CLI usage examples, including (i) data downloading in Appendix E.1, (ii) data processing in Appendix E.2, (iii) model training in Appendix E.3, (iv) model inference in Appendix E.4, and (v) biological interpretation in Appendix E.5. Then we conclude with the WebUI demonstration in Appendix E.6.

Genome-Factory relies on YAML configuration files to define tasks, with example files available in *"genomeFactory/Examples"*. Users can customize the parameters in these files as long as the required YAML structure is preserved.

## E.1  DATA DOWNLOADING

Users can download data from NCBI. It supports downloading from a config file or an interface.

**Using a Config File.**  Specify download parameters in a YAML file. It supports both species-based and link-based downloads.

- Download by species.

```
genomefactory-cli download genomeFactory/Examples/
    download_by_species.yaml
```

- Download by Link.

```
genomefactory-cli download genomeFactory/Examples/
    download_by_link.yaml
```

**Using the Interactive Interface.**  Run the command without a config file and follow the prompts to specify your download criteria (this supports both species-based and link-based downloads).

```
genomefactory-cli download
```

For both two methods, the list of species and their taxonomy IDs used for downloads is stored in *"genomeFactory/Data/Download/Datasets_species_taxonid_dict.json"*. Users can extend it by adding new species–taxonomy ID pairs to enable downloads for additional species.

## E.2  DATA PROCESSING

GENOME-FACTORY provides tools to prepare data for model fine-tuning. This includes (i) processing data downloaded from NCBI (Appendix E.2.1), (ii) formatting users' own custom datasets (Appendix E.2.2), and (iii) processing specialized genomic data (Appendix E.2.3).

### E.2.1  PROCESSING NCBI DATA

Follow the steps below to process the data downloaded from NCBI.

- Gather data downloaded into a single folder (steps in Appendix E.1 finish this automatically).

- Run the processing command with a config file.

```
genomefactory-cli process genomeFactory/Examples/process_normal.
    yaml
```

- The processed data will be ready for model fine-tuning.

### E.2.2  PREPARING CUSTOM DATASETS

Follow the steps below to process the custom data.

- Separate the data into three CSV files: *"train.csv"*, *"dev.csv"*, and *"test.csv"*.

- Each CSV file must have two columns: (i) The first column should contain the DNA sequences (e.g., *"sequence"*). (ii) The second column should contain the corresponding labels (e.g., *"label"*). For classification tasks, labels should be integers (e.g., 0, 1, 2). For regression tasks, labels should be continuous numbers.

- Place these three CSV files (*"train.csv"*, *"dev.csv"*, and *"test.csv"*) together in a single folder.

- Specify this folder as the input data directory in your training configuration YAML file.

### E.2.3  SPECIALIZED PROCESSING

GENOME-FACTORY provides specialized dataset generation tools for common genomic tasks.

- Promoter region dataset: Generate promoter vs. non-promoter classification data for human, mouse, and zebrafish genomes.

```
genomefactory-cli process genomeFactory/Examples/
    process_promoter.yaml
```

- Epigenetic mark dataset: Create gene body sequences with H3K36me3 signal classification for human and mouse genomes.

```
genomefactory-cli process genomeFactory/Examples/process_emp.
    yaml
```

- Enhancer region dataset: Build enhancer vs. non-enhancer classification data for human and mouse genomes.

```
genomefactory-cli process genomeFactory/Examples/
    process_enhancer.yaml
```

All three datasets feature quality control, configurable train/val/test splits, and output CSV files with *"sequence", "label"* format.

### E.3  MODEL TRAINING

To fine-tune GFMs, GENOME-FACTORY supports two primary task types: classification and regression. Users can specify the desired *task_type* in the training YAML configuration file.

Follow the steps below to fine-tune GFMs using three different methods.

**Full-parameter Fine-tuning.** The following command applies to all models except for EVO. We do not support the full-parameter fine-tuning for EVO now.

```
genomefactory-cli train genomeFactory/Examples/train_full.yaml
```

**Low-rank Adaptation Fine-tuning (LoRA).**

- For general models (except for EVO), users can use:

```
genomefactory-cli train genomeFactory/Examples/train_lora.yaml
```

- For the EVO model, users need to use:

```
genomefactory-cli train genomeFactory/Examples/train_evo_lora.
    yaml
```

For all models, users can specify target modules in the YAML file: (i) *"all"* - targets all linear layers; (ii) *"all_in_and_out_proj"* - targets input/output projection linear layers and the final classification layer; and (iii) *"custom"* - specifies module names by users.

**Adapter-based Fine-tuning.** The following command applies to all models.

```
genomefactory-cli train genomeFactory/Examples/train_adapter.yaml
```

In *"genomeFactory/Train/workflow/adapter/adapter_model/Adapter.py"*, users can customize the adapter architecture for better performance on specific downstream tasks.

For all three methods, training settings (e.g., batch size, learning rate, and training epochs) can be customized in the respective YAML files.

**Benchmarking:** After fine-tuning, performance metrics are saved to a JSON file. Users can use these metrics for benchmarking (e.g., comparing the performance of different models or tuning methods).

### E.4  MODEL INFERENCE

Follow the steps below to use genomic models for embedding extraction and generation.

**Embedding Extraction.** Extract the last hidden state embeddings from sequences.

- For the general case (except for EVO):

```
genomefactory-cli inference genomeFactory/Examples/
    inference_extract.yaml
```

- For EVO:

```
genomefactory-cli inference genomeFactory/Examples/
    inference_extract_evo.yaml
```

**Generation.** Generate new DNA sequences based on prompts. Applicable to generative models.

- For GenomeOcean:

```
genomefactory-cli inference genomeFactory/Examples/
    inference_generation_genomeocean.yaml
```

- For EVO:

```
genomefactory-cli inference genomeFactory/Examples/
    inference_generation_evo.yaml
```

### E.5  BIOLOGICAL INTERPRETATION

GENOME-FACTORY provides comprehensive tools for understanding and interpreting GFMs through a sparse auto-encoder (SAE) to provide biological insights into model behavior.

Follow the steps below to implement the workflow for interpreting.

**Train an SAE Model.** Use the following command to train SAE with the DNA sequences.

```
genomefactory-cli sae_train genomeFactory/Examples/sae_train.yaml
```

Configure the following parameters in the YAML file. Users need to specify the DNA sequence file as *data_file*. Use *model_name* to assign a name or checkpoint path for the genomic model. Set *d_model* to the output dimension of the genomic model used to generate the embeddings. Define the SAE bottleneck dimension using *d_hidden*. It controls the number of sparse latent features.

```
data_file: "<YOUR_DNA_SEQUENCE_FILE>"
d_model: "<GENOMIC_MODEL_OUTPUT_DIMENSION>"
d_hidden: "<SAE_HIDDEN_DIMENSION>"
model_name: "<MODEL_NAME_AND_PATH>"
```

**Downstream Evaluations with Ridge Regression.** Users need to generate hidden embeddings from genomic models and save the results as a CSV file at the path specified by *"<GE-NOMIC_MODEL_FEATURE_CSV_PATH>"* (by Appendix E.4). Then use [CLS] tokens embedding or mean-pooling embedding to do further analysis.

- Use the [CLS] token latent embedding for analysis.

```
genomefactory-cli sae_regression genomeFactory/Examples/
    sae_regression.yaml
```

Configure the following parameters in the YAML file. Users need to specify the DNA sequence embedding file as *csv_path*. Use *sae_checkpoint_path* to assign the checkpoint path of the SAE.

```
csv_path: "<GENOMIC_MODEL_FEATURE_CSV_PATH>"
sae_checkpoint_path: "<SAE_CHECKPOINT_PATH>"
type: "<first_token>"
```

- Use the mean-pooling latent embedding for analysis.

```
genomefactory-cli sae_regression genomeFactory/Examples/
    sae_regression.yaml
```

Configure the following parameters in the YAML file.

```
csv_path: "<GENOMIC_MODEL_FEATURE_CSV_PATH>"
sae_checkpoint_path: "<SAE_CHECKPOINT_PATH>"
type: "<mean>"
```

### E.6  USAGE VIA WEBUI

Use the following command to access all GENOME-FACTORY functionalities by a graphical interface.

```
genomefactory-cli webui
```

This command launches a web server. Open the provided URL in users' browsers to use the WebUI.

