# OpenReview forum: "Genome-Factory: A Library for Tuning, Deploying, and Interpreting Genomic Models"
_ICLR.cc/2026/Conference — ICLR 2026 Conference Withdrawn Submission_

### Official Review · Reviewer_Hedo · 2025-10-26

**Soundness:** 3
**Presentation:** 3
**Contribution:** 2
**Rating:** 4
**Confidence:** 1

**Summary:**

This paper introduces GENOME-FACTORY, a Python library designed to unify the workflow for genomic foundation models (GFMs). The framework integrates six components: (1) Genome Collector for data acquisition and preprocessing, (2) Model Loader supporting diverse genomic models (HyenaDNA, DNABERT-2, Caduceus, Nucleotide Transformer, EVO, GenomeOcean), (3) Model Trainer enabling full-parameter, LoRA, and adapter-based fine-tuning, (4) Inference Engine for embedding extraction and sequence generation, (5) Benchmarker providing evaluation on GUE and Genomic Benchmarks, and (6) Biological Interpreter using sparse autoencoders (SAE) for interpretability. The authors validate their framework across three dimensions: compatibility with diverse models, benchmarking performance, and biological interpretation using DNABERT-2.

**Strengths:**

* Comprehensive Integration: The paper addresses a practical need by consolidating various components of the genomic model workflow into a single framework. The integration of data preprocessing, model training, inference, and evaluation is useful for practitioners.
* Broad Model Support: Supporting six different genomic foundation models with varying architectures (attention-based, state-space models, generative models) demonstrates reasonable engineering effort and provides users with flexibility.
* Multiple Fine-tuning Strategies: The inclusion of full-parameter, LoRA, and adapter-based tuning with detailed efficiency analysis (Table 2, Table 3) provides actionable insights about computational trade-offs.
* User-Friendly Interfaces: Offering both CLI and WebUI lowers the barrier to entry for non-expert users, which could facilitate broader adoption in the biology community.
* Systematic Benchmarking: The evaluation across multiple models and fine-tuning methods on two benchmark suites (GUE and Genomic Benchmarks) provides comprehensive performance comparisons.

**Weaknesses:**

* Limited Research Novelty: The paper primarily presents an engineering integration effort rather than novel research contributions suitable for ICLR. While the authors claim to introduce "the first integrated Python library" for genomic models, this represents an engineering milestone rather than a fundamental research advance. Each component relies on existing techniques without significant innovation: data preprocessing employs standard bioinformatics operations, fine-tuning methods are direct applications of LoRA and adapters from NLP literature, and the interpretability module uses standard sparse autoencoders without genomics-specific innovations. The primary contribution is integrating these existing components into a unified interface, which, while potentially useful for practitioners, does not constitute the level of methodological or scientific novelty expected at ICLR.
* Weak Biological Interpretation Validation: The biological interpretation component (Section 4.3), positioned as a key contribution and claimed to provide interpretability for genomic foundation models, suffers from severe validation weaknesses. The interpretability experiment only examines 2,000 sequences from two species, which is insufficient for generalizable conclusions. The discovered features (units 382, 519, and 3519 correlating with sequence length) represent trivial patterns that any embedding would naturally capture. The paper provides no evidence of monosemanticity, lacking feature visualization, maximally activating examples, or intervention experiments demonstrating causal relationships. There is no biological validation: no comparison with established motif databases like JASPAR, no experimental verification, and no domain expert evaluation. The regression approach only demonstrates correlation, not causation or biological mechanism. The claim of providing meaningful biological interpretation is substantially overstated.
* Incomplete Benchmarking and Evaluation: The experimental evaluation contains several significant gaps that limit the conclusions. The benchmarking lacks comparisons with traditional ML baselines (SVMs, Random Forests on k-mer features, simple CNNs). While the paper focuses on comparing foundation models, including traditional baselines would help establish when the additional computational overhead is justified for specific genomic tasks.
* Unclear Problem Motivation: The paper's motivation, while addressing a practical need, is not sufficiently established for a research venue. The claimed "fundamental gap" between biologists and engineers is not substantiated with user studies, surveys, or concrete failure cases that would demonstrate this is a research problem rather than purely an engineering convenience. Existing tools (Biopython, scikit-bio, HuggingFace genomics models) already address many mentioned pain points. Unlike LLaMA-Factory, which addressed severe pain points in LLM fine-tuning (distributed training, memory management), the paper does not convincingly demonstrate that genomic model deployment faces comparable challenges requiring a new unified framework.

**Questions:**

* Regarding Biological Interpretation (Section 4.3): Can you provide evidence that discovered features are monosemantic beyond correlation with sequence length? Have these features been validated against known biological elements such as motifs or regulatory sites?
* Regarding Benchmarking: Why is comparison with traditional ML baselines (SVM, Random Forest on k-mer features, simple CNNs) missing from your evaluation? What is the performance-cost trade-off compared to simpler baselines, and under what conditions are foundation models actually necessary versus traditional approaches?
* Regarding User Needs and Problem Motivation: Can you provide user studies or feedback from biologists demonstrating the claimed gap between domain expertise and technical implementation? What specific use cases cannot be adequately addressed by existing tools such as the combination of HuggingFace genomics models and Biopython?

---

### Official Review · Reviewer_GnpM · 2025-10-29

**Soundness:** 3
**Presentation:** 2
**Contribution:** 2
**Rating:** 4
**Confidence:** 4

**Summary:**

The paper presents a Python library that unifies the pipeline of using genomic models. It integrates 6 components into this framework, which are data collector, model loader, model trainer, inference engine, benchmarker, and biological interpreter. The main contribution is creating an integrated platform that bridges the gap between biological and engineering experts.

**Strengths:**

The paper proposes a plausible solution to a clear need in the domain of large genomic models. The framework and interface design are user-friendly and support many genomic models, which is valuable for the community. The paper evaluates the framework across different settings with concrete benchmarks and metrics.

**Weaknesses:**

1. Although it’s meaningful to integrate multiple large genomic models, the paper’s novelty is primarily in the engineering part, and the technical novelty is limited. There are no clear new biological or methodological insights from the benchmarking results.

2. The benchmarking is limited. They only compared the models on the classification tasks and didn’t include regression tasks.

3. Section 4.3 on biological interpretation is underexplored. Only sequence length is used as the biological feature for interpretation validation. After they identified the important latent features, it lacked evaluation to prove that the interpretation method is indeed accurate.

4. Although the paper mentions how to use the framework through the WebUI in Section E.6, it lacks details about the implementation and does not show what the webpage looks like.

**Questions:**

1. Why did the authors choose the sparse auto encoder as the biological interpreter over other interpretation methods?
2. Could you provide case studies showing how biologists who lack coding expertise used the WebUI?

---

### Official Review · Reviewer_bfzp · 2025-10-31

**Soundness:** 3
**Presentation:** 3
**Contribution:** 1
**Rating:** 2
**Confidence:** 4

**Summary:**

This paper presents GENOME-FACTORY, a unified Python library for genomic foundation models. It streamlines the full pipeline - data collection, fine-tuning, inference, benchmarking, and interpretability under a single roof.

**Strengths:**

1. Provides a single pipeline from data acquisition to model interpretation.
2. Each module from Genome Collector to Biological Interpreter  is clearly structured, facilitating reproducibility and extensibility.

**Weaknesses:**

1. Many core functions (model loading, LoRA/adapters, benchmarking) reuse Hugging Face + Trainer extensively; novelty is lacking.
2. Beyond sparse-autoencoder interpretability, there is limited methodological innovation
3. The biological interpretation experiment relies on sequence length regression, which is not a biologically meaningful property. Demonstrating neuron-length correlation does not prove the model captures biological mechanisms

**Questions:**

My suggestions is that this paper would be more suitable for a Datasets & Benchmarks track at some conferences or a workshop rather than the main track of the conference.

---

### Official Review · Reviewer_ZdrW · 2025-11-01

**Soundness:** 2
**Presentation:** 2
**Contribution:** 2
**Rating:** 4
**Confidence:** 5

**Summary:**

This paper presents GENOME-FACTORY, a comprehensive Python library designed to unify the tuning, deployment, and interpretation of genomic foundation models (GFMs). The system integrates six major components—Genome Collector, Model Loader, Model Trainer, Inference Engine, Benchmarker, and Biological Interpreter—providing an end-to-end workflow for genomic AI. The authors claim the framework bridges the gap between model developers and domain biologists by combining (i) multi-paradigm fine-tuning (full, LoRA, adapter), (ii) standardized evaluation via existing genomic benchmarks, and (iii) interpretability through sparse auto-encoders for biological feature mapping. Experiments benchmark several existing GFMs (DNABERT-2, HyenaDNA, Caduceus, EVO, GenomeOcean) and provide results on efficiency, throughput, and limited biological interpretation.

**Strengths:**

- **(S1)** System-Level Integration Filling a Practical Gap. The framework addresses a critical infrastructural bottleneck in genomics: the lack of unified pipelines for fine-tuning and evaluating GFMs. This integration—while not algorithmically novel—offers clear practical value by connecting data processing, model training, inference, and biological interpretation in a consistent ecosystem. The modular design is logically sound, extensible, and lowers the engineering barrier for practitioners.

- **(S2)** Experimental Breadth and Practical Relevance. The empirical section, though not deeply analytical, effectively demonstrates the framework’s compatibility and efficiency across several major GFMs (DNABERT-2, HyenaDNA, Caduceus, Nucleotide Transformer, EVO, GenomeOcean). The results convincingly show trade-offs between resource consumption and accuracy across tuning paradigms, validating the framework’s real-world applicability.

- **(S3)** Writing Quality and Presentation Clarity. The manuscript is well organized and written in clear, professional English. Figures (notably Fig. 1 and Fig. 2) are well labeled and visually intuitive, providing effective overviews of the system pipeline and experimental trade-offs. The technical descriptions are coherent and self-contained.

**Weaknesses:**

- **(W1) Insufficient Conceptual and Algorithmic Novelty.** The paper is mainly a software integration effort rather than a methodological advance. No new model, optimization technique, or theoretical insight is introduced. The sparse auto-encoder–based interpretability is neither theoretically justified nor benchmarked against existing genomic interpretability methods such as gradient-based attribution, motif visualization, or probing. As a result, the contribution is more infrastructural than scientific, limiting its novelty.

- **(W2) Lack of Biological Relevance in Validation.** The interpretability demonstrations focus on trivial correlations (e.g., sequence length), which provide little biological insight. There is no analysis on whether the learned latent dimensions align with meaningful biological patterns such as transcription factor motifs, enhancer-promoter relations, or epigenetic states. This weakens the core claim that GENOME-FACTORY “interprets” genomic models.

- **(W3) Limited Benchmark Scope and Missing State-of-the-Art Models.** The evaluation primarily relies on GUE and Genomic Benchmarks, which cover only low-complexity classification tasks. The authors omit key state-of-the-art models (e.g., GROVER [1], GUANinE v1.0 [2], LRB [3]) and multi-species data. This omission constrains both the fairness and generalizability of the reported findings.

- **(W4) Weak Empirical Analysis and Missing Ablations.** The study does not provide ablation or sensitivity experiments for adapter width, LoRA rank, or sparsity coefficients, which are central to the proposed framework’s efficiency claims. Furthermore, the interpretability module’s hyper-parameters (e.g., sparsity penalties, latent dimensions) are not justified, preventing a rigorous understanding of its behavior.

- **(W5) Incomplete Documentation and Reproducibility Evidence.** The reproducibility claim is not fully verifiable. While the paper claims reproducibility via released code and YAML configurations, the main text lacks sufficient implementation transparency—for example, data versioning, preprocessing scripts, or computational cost breakdowns. Meanwhile, the paper does not specify dataset versions, preprocessing scripts, or exact software dependencies. Moreover, hardware specifications, GPU hours, and FLOPs comparisons are missing, which prevents fair efficiency assessment.

### Reference

[1] The human genome’s vocabulary as proposed by the DNA language model GROVER. bioRxiv, 2023.

[2] GUANinE v1.0: Benchmark Datasets for Genomic AI Sequence-to-Function Models. bioRxiv, 2023.

[3] The Genomics Long-Range Benchmark: Advancing DNA Language Models. OpenReview, 2025.

**Questions:**

- **(Q1)** How does the sparse auto-encoder compare quantitatively to attribution-based interpretability techniques (e.g., Integrated Gradients, DeepLIFT, SHAP) in identifying biologically meaningful features?

- **(Q2)** Could the authors include recent GFMs (e.g., DNABERT-S, Evo-2) in the benchmark to strengthen the fairness and relevance of the evaluation?

- **(Q3)** How generalizable is GENOME-FACTORY to multi-species datasets or 3D genome tasks (e.g., Hi-C, enhancer-gene prediction)?

- **(Q4)** Can the interpretability component be integrated with other forms of biological visualization (e.g., motif heatmaps, attention aggregation plots)?

---

### Note · Authors · 2025-11-26

**Comment:**

We would like to thank the reviewers for their valuable feedback and constructive comments on our submission. After careful consideration, we have decided to withdraw our paper at this stage.

**Withdrawal Confirmation:**

I have read and agree with the venue's withdrawal policy on behalf of myself and my co-authors.